# Examining the predictive accuracy of metabolomics for small-for-gestational-age babies: a systematic review

Debora Farias Batista Leite,[1,2] Aude-Claire Morillon,[3] Elias F Melo Júnior,[4] Renato T Souza,[5] Fergus P McCarthy,[6] Ali Khashan,[7] Philip Baker,[8] Louise C Kenny,[9] Jose Guilherme Cecatti[10]

**Correspondence to**
Professor Jose Guilherme Cecatti; cecatti@unicamp.br

## ABSTRACT

**Introduction** To date, there is no robust enough test to predict small-for-gestational-age (SGA) infants, who are at increased lifelong risk of morbidity and mortality.

**Objective** To determine the accuracy of metabolomics in predicting SGA babies and elucidate which metabolites are predictive of this condition.

**Data sources** Two independent researchers explored 11 electronic databases and grey literature in February 2018 and November 2018, covering publications from 1998 to 2018. Both researchers performed data extraction and quality assessment independently. A third researcher resolved discrepancies.

**Study eligibility criteria** Cohort or nested case–control studies were included which investigated pregnant women and performed metabolomics analysis to evaluate SGA infants. The primary outcome was birth weight <10th centile—as a surrogate for fetal growth restriction—by population-based or customised charts.

**Study appraisal and synthesis methods** Two independent researchers extracted data on study design, obstetric variables and sampling, metabolomics technique, chemical class of metabolites, and prediction accuracy measures. Authors were contacted to provide additional data when necessary.

**Results** A total of 9181 references were retrieved. Of these, 273 were duplicate, 8760 were removed by title or abstract, and 133 were excluded by full-text content. Thus, 15 studies were included. Only two studies used the fifth centile as a cut-off, and most reports sampled second-trimester pregnant women. Liquid chromatography coupled to mass spectrometry was the most common metabolomics approach. Untargeted studies in the second trimester provided the largest number of predictive metabolites, using maternal blood or hair. Fatty acids, phosphosphingolipids and amino acids were the most prevalent predictive chemical subclasses.

**Conclusions and implications** Significant heterogeneity of participant characteristics and methods employed among studies precluded a meta-analysis. Compounds related to lipid metabolism should be validated up to the second trimester in different settings.

**PROSPERO registration number** CRD42018089985.

## Strengths and limitations of this study

► To our knowledge, this is the first systematic review to assess the predictive accuracy of metabolomics for an adverse pregnancy outcome.

► Using small for gestational age (SGA) as surrogate for fetal growth restriction—just as in epidemiological investigations—improves the translational potential of metabolomics.

► Identification of techniques, types of maternal samples and chemical classes paves the way for future metabolomics investigations on fetal growth patterns.

► Available data could not support a meta-analysis; further studies should include accuracy measures of individual metabolites or chemical subclasses in predicting SGA.

## INTRODUCTION

Fetal growth restriction (FGR) and small-for-gestational-age (SGA) infants are major concerns in modern obstetrics.[1–3] SGA is commonly used as a proxy for FGR,[4] despite the subtle differences between these two pathological conditions. The prevalence of both varies according to the criteria applied and on the population and setting, although it reaches as much as 25% in low-income and middle-income countries.[5] SGA newborns may have adverse health effects, such as stillbirth,[4] perinatal asphyxia,[6] impaired neurodevelopment[7] and increased cardiovascular risk.[8 9] To date, there are no robust prediction tools for SGA using clinical factors,[10 11] ultrasound data[12 13] or placental biomarkers.[14]

For hypothesis-generating or validation purposes, metabolomics is a novel area of biomarker, discovery, development and clinical diagnostics in translational medicine.[15 16] Metabolomics is the study of all metabolites[15 16] in a given sample, that is, low molecular weight compounds (50–2000 Da) that are intermediates of biochemical reactions and metabolic pathways, considered to directly reflect cellular activity and phenotype.[15 16] Recent studies have evaluated the pathophysiology[17–20] of SGA with

metabolomics. However, little is known about the potential of metabolomics to identify predictive compounds of SGA.

Since metabolomics can identify multiple metabolites from low volume samples and create a model from a collection of these samples,[15] it is a promising technology for hypothesis generation in a heterogeneous condition such as SGA. The prediction of SGA in pregnancy would help refer women to specialised care facilities, improving maternal and neonatal outcomes.[21 22]

In this context, our review question was 'What is the accuracy of metabolomics for predicting FGR?'. The main objective of this systematic review was to assess the accuracy of metabolomics techniques in predicting SGA. As a secondary aim, we intended to determine which metabolites are predictive of this condition.

## METHODS

The protocol for this systematic review was published previously.[23] This study follows international guidelines for transparency (International Prospective Register of Systematic Reviews) and respects the Preferred Reporting Items for Systematic Reviews and Meta-Analyses statement.[24]

### Literature search strategy

Two independent researchers (DFBL and A-CM) assessed 11 electronic databases (PubMed, EMBASE, Latin American and Caribbean Health Sciences Literature, Scientific Electronic Library Online, Health Technology Assessment, Database of Abstracts of Reviews of Effects, Aggressive Research Intelligence Facility, Cumulative Index of Nursing and Allied Health Literature, Maternity and Infant Care, Scopus, and Web of Science) and grey literature. There were no limits or language constraints; the search strategy covered published documents between 1998 and 2018. The keywords 'small for gestational age', 'metabolomics', 'prediction' and 'antenatal', and variations of each, were combined with Boolean operators depending on each database requirements. The full EMBASE literature search was as follows: ('fetal growth retardation' OR 'fetal growth restriction' OR 'intrauterine growth restriction' OR 'intrauterine growth retardation' OR 'small for gestational age') AND ('metabolomic*' OR 'metabonomic*' OR 'metabolit* 'H NMR' OR 'proton NMR' OR 'proton nuclear magnetic resonance' OR 'liquid chromatogra*' OR 'gas chromatogra*' OR 'UPLC' OR 'ultra-performance' OR 'ultra performance liquid chromatograph*') AND ('pregnan*' OR 'antenat*' OR 'ante nat*' OR 'prenat*' OR 'pre nat*') AND ('screen*' OR 'predict*' OR 'metabolic profil*'). Please check online supplementary material 1 for more details.

### Outcomes and subgroup analysis

The primary outcome was SGA, as a surrogate for FGR and defined as birth weight <10th centile, by population-based or customised charts. The secondary outcomes were birth weight ≤5th or ≤3rd centile.

The intended subgroup analysis comprised the type of metabolomics technique applied (nuclear magnetic resonance, NMR; gas or liquid chromatography coupled with mass spectrometry, GC-MS or LC-MS, respectively); maternal health status before pregnancy (women with vs without any chronic health condition); type of SGA suspected during pregnancy (early vs late SGA); and type of pregnancy (singleton vs multiple pregnancy).

### Selection criteria of studies, data collection and analysis

Cohort or case–control studies were included if maternal samples were collected before the clinical diagnosis of SGA, if any metabolomics technique was applied and if the results of SGA were presented. Articles presenting data from the same research project but analysing distinct metabolites or showing data from different countries were included. Studies were excluded (1) according to study design; (2) if they had not applied any metabolomics technique; (3) if they were only experimental studies; (4) if it was not possible to extract data on SGA; or (5) if they presented duplicate data, in which case the most complete publication was included for final analysis.

Two researchers (DFBL and A-CM) independently selected studies, extracted data and discussed discrepancies. One additional reviewer (EFMJ or RTS) helped to decide, by majority, when no consensus was reached.

Piloted standardised forms were applied for data extraction, including pregnancy characteristics and experimental details. The Human Metabolome Database (HMDB)[25] and the Kyoto Encyclopedia of Genes and Genomes[26] were used for matching chemical class and metabolic pathways of each metabolite, respectively.

### Risk of bias and assessment of concerns regarding applicability

Two researchers (DFBL and A-CM) independently evaluated individual studies using the Quality Assessment of Diagnostic Accuracy Studies-2 (QUADAS-2) tool.[27] One of the third reviewers (EFMJ or RTS) helped in decision-making when no consensus was achieved.

Each study was classified as high, low or unclear risk of bias in four domains (patient selection, index test, reference standard, and flow and timing), and as high, low or unclear concerns regarding applicability in the first three domains. We did not consider two signalling questions ('Was a case-control design avoided?' and 'Was there an appropriate interval between the index test and reference standard?'). The nested case–control design was an inclusion criterion, and maternal samples should have been collected during pregnancy, that is, before the SGA diagnosis. Studies were considered 'low risk', for example, (1) if pregnancy or neonatal complications were not excluded in just one group of participants or data on participant selection had been provided; (2) if methods for sample preparation and interpretation were standardised or metabolite threshold was defined before the experiments

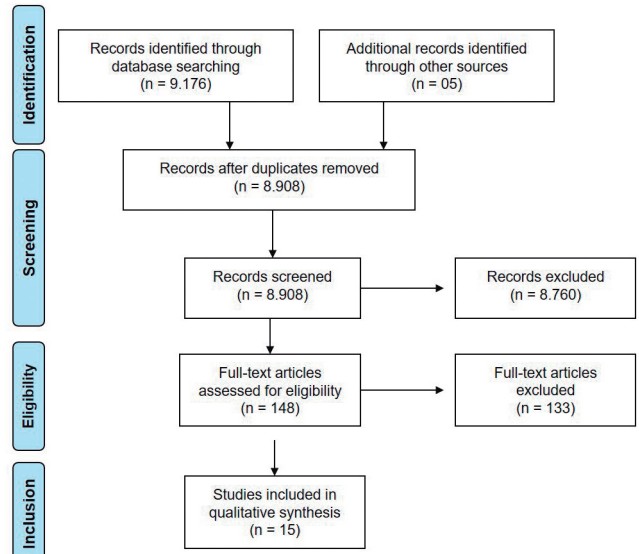

**Figure 1** PRISMA flow chart of study identification, screening and selection. PRISMA, Preferred Reporting Items for Systematic Reviews and Meta-Analyses. From Moher D, *et al*[24] For more information, visit www.prisma-statement.org.

(for targeted analysis); (3) if the adequacy and reasons for choosing the reference birthweight chart had been explained; or (4) if large-for-gestational-age babies had been excluded from the final comparative analysis.

### Data synthesis
A quantitative summary of data was performed when any predictive accuracy measures could be extracted. Authors were contacted to provide additional information, when necessary. However, only Delplancke *et al*[28] replied. The estimation of likelihood ratios and hierarchical summary receiver operator characteristic curve[29] was planned, as well as assessment of heterogeneity and publication bias.[30] However, due to lack of data, a meta-analysis could not be performed.

### Patient and public involvement
There was no patient or public involvement in conducting this systematic review.

### RESULTS
#### Literature search characteristics
The literature search for this systematic review was performed in February 2018 and rerun in November 2018. A total of 9181 references were retrieved (figure 1). After the removal of duplicate records (n=273), title and abstract screening, and analysis of the remaining 148 full-text articles, 15 articles were included.[17 18 28 31–42] See online supplementary material 2 for the excluded studies.

#### Characteristics of the included studies
The characteristics of the included studies are shown in table 1. The prevalence of SGA ranged from 7.3%[33] to 21.5% in cohort studies.[28] There were no studies using birth weight ≤3rd centile to define SGA. The time interval

between initial participant enrolment and publication varied from 3[17] to 54 years,[40] although these data were unclear in 38% of the reports.[18 28 32 33 37] In nested case–control studies, participants were matched by maternal age,[17 18 38 42] ethnicity,[17 18 42] parity,[38] body mass index[17 18 42] or infant gender.[18 38]

Participant characteristics varied between studies. Regarding gestational age at assessment, samples were collected in the second trimester in half of the studies.[17 18 33 35 37 39 42] In three reports, women were assessed at least twice.[34 38 41] In one study, maternal blood was drawn either in the first or second trimester,[40] and in another three studies only samples from the third trimester were considered.[28 36 41] In the latter case, maternal hair was divided according to length, allowing evaluation of second-trimester and third-trimester metabolites.[28] Studies considering the fifth centile as the cut-off sampled women in the first trimester.[31 32] Twin pregnancy was a clear exclusion criterion in most studies.[17 18 31 33–35 37 40–42] Pregnancy aided by assisted reproduction[18 37] or women with pre-existing conditions[17 18 35 37 42] were also excluded, although these data were incompletely reported.[28 32 36 38 39 41] When both nulliparous and multiparous women were enrolled, there was no data analysis according to parity. Half of the studies considered term deliveries exclusively,[18 28 36 38–41] and the remaining studies did not differentiate results according to gestational age at birth.

Regarding clinical risk factors for SGA, only one paper mentioned a history of SGA, but findings were not adjusted for this variable.[32] All studies, except one,[28] cited participants' smoking status. The rate of smoking habit ranged from 2.4%[18] to 47.5%.[40] It is important to note that Gernand *et al*[40] analysed samples from women recruited between 1959 and 1965, when smoking while pregnant was encouraged, which explains the high rate of smoking participants. The duration of smoking or any differences in birth weight (absolute measures or centiles) were not clearly stated. Although more prevalent in SGA pregnancies, the results did not change with this variable control.[31 32 35 37 40] Only Gong *et al*[41] mentioned the suspicion of SGA in pregnancy, exhibiting decreasing abdominal circumference growth velocity between 20 and 36 weeks. However, on final analysis, these babies were grouped with infants not suspected during pregnancy.

#### Subgroup analysis
Due to unavailable data, the only subgroup analysis performed was related to the metabolomics approach applied (table 2). There was no mention of adherence to metabolomics reporting data guidelines. LC-MS was the leading technique used. Three studies have investigated metabolites related to environmental exposure, from contaminated water,[31] consumer products[36] or pesticides,[42] while others have analysed endogenous compounds.[32–35 37–40] Only Luthra *et al*[38] conducted a biomarker validation study, while Gong *et al*[41] chose

**Table 1** Main characteristics of included studies

| Authors, year | Country, year of participants' enrolment | Study design | Affected/Non-affected | Gestational age at assessment | Type of pregnancy | Parity | Birthweight curve |
|---|---|---|---|---|---|---|---|
| Outcome: SGA <5th centile | | | | | | | |
| Costet et al, 2012[31] | France, 2002–2006 (PELAGIE cohort) | Nested case–control | 134/399 | 11 weeks | Single pregnancy | Nulliparous and parous women, unclear proportions | Customised curve |
| Ertl et al, 2012[32] | UK* | Nested case–control | 150/1000 | 11^{+0}–13^{+6} weeks | Unclear | 55.3% nulliparous in SGA group, 48.1% nulliparous in control group | Population-based charts |
| Outcome: SGA <10th centile | | | | | | | |
| Grandone et al, 2006[33] | Italy* | Cohort | 31/393 | 17.1±1.2 weeks† (mean) | Single pregnancy; no maternal pre-existing conditions | Unclear | Population-based charts |
| van Eijsden et al, 2008[39] | The Netherlands, 2003–2004 (ABCD study) | Cohort | 429/3275 | 13.5±3.3 weeks (mean) | Term deliveries, no diabetes or hypertension | 57.6% nulliparous | Population-based charts |
| Horgan et al, 2011[17] | Australia, 2008–2011 (SCOPE cohort) | Nested case–control | 40/40 | 14–16 weeks | Single pregnancy; no other pregnancy complications | Nulliparous | Customised curve |
| Gernand et al, 2013[40] | USA, 1959–1965 (Collaborative Perinatal Project) | Nested case–control | 395/1751 | ≤26 weeks | Single pregnancy; term deliveries | Parous women | Population-based charts |
| Sulek et al, 2014[18] | Singapore* (GUSTO study) | Nested case–control | 41/42 | 26–28 weeks | Single pregnancy; term deliveries; no maternal pre-existing conditions | Nulliparous and parous women, unclear proportions | Population-based charts |
| Choi et al, 2016[34] | South Korea, 2012–2013 | Cohort | 39/217 | First, second or third trimester | Single pregnancies | Nulliparous and parous women, unclear proportions | Population-based charts |
| Kiely et al, 2016[35] | Ireland, 2008–2011 (SCOPE cohort) | Cohort | 190/1578 | 14–16 weeks | Single pregnancy; no maternal pre-existing conditions | Nulliparous | Customised curve |
| Ong et al, 2016[37] | Singapore* (GUSTO study) | Cohort | 83/827 | 26–28 weeks | Single pregnancy; no maternal chronic illness | 43.5% nulliparous | Population-based charts |
| Wang et al, 2016[36] | Taiwan, 2000–2001 (Taiwan Maternal and Infant Cohort Study) | Cohort | 35/188 | Third trimester | Unclear; term deliveries | 48% nulliparous | Population-based charts |
| Delplancke et al, 2018[28] | New Zealand* | Cohort | 20/73 | 34–37 weeks | Unclear; term deliveries | Unclear | Customised curve |
| Luthra et al, 2018[38] | USA, 2010–2012 (TIDES study) | Nested case–control | 53/106 | First and second trimester | Single pregnancies; term deliveries | 60% nulliparous | Customised curve |

Continued

| Authors, year | Country, year of participants' enrolment | Study design | Affected/Non-affected | Gestational age at assessment | Type of pregnancy | Parity | Birthweight curve |
|---|---|---|---|---|---|---|---|
| Gong et al, 2018[41] | UK, 2008–2012 (POP study) | Nested case–control | 162/259 | 36 weeks | Single pregnancies; term deliveries | Nulliparous | Customised curve |
| Morillon et al, 2018[42] | 2008–2011 (SCOPE study) | Nested case–control | 40/40 | 20 weeks | Single pregnancies | Nulliparous | Customised curve |

Table 1  Continued

*Unclear period of participant recruitment.
†Mean for all study participants.
ABCD, Adolescent Brain Cognitive Development; GUSTO, Growing Up in Singapore Towards healthy Outcomes; PELAGIE, Étude Longitudinale sur les Anomalies de la Grossesse, l'Infertilité et l'Enfance; POP, Pregnancy Outcome Prediction; SCOPE, Screening of Pregnancy Endpoints; SGA, small for gestational age; TIDES, Tackling Inequalities and Discrimination Experiences in health Services.

to analyse the top 10 statistically different metabolites according to infant sex.

Maternal blood was the most common biological sample analysed by LC-MS in all studies,[17 32 34–37 39–41] except for one which used GC-MS.[39] Maternal urine was analysed by NMR,[38] GC-MS[36] or LC-MS.[42] There was only one report using amniotic fluid[33] and two using maternal hair,[18 28] all applying GC-MS. The period of laboratory analysis was rarely specified, which made it impossible to estimate the total time of sample storage.

Untargeted studies reported diverse metabolic features. Authors matched the peaks with an inhouse library[18 28] or HMDB-related database.[17 42] Horgan et al[17] found 785 compounds both in maternal and newborn samples; their predictive model included 19 metabolites (only 5 could be putatively identified; table 2) and used second-trimester maternal blood. Sulek et al[18] and Delplancke et al[28] prepared and analysed samples with GC-MS using similar protocols. Sulek et al[18] identified 32 statistically different chromatographic features from which they built a predictive model using five metabolites, including two fatty acids (2-methyloctadecanoate and margarate). In contrast, Delplancke et al[28] identified 198 metabolites, including three fatty acids (margaric, pentadecanoic and myristic acid) showing significantly higher levels in SGA cases, when second-trimester maternal hair segments were studied.

### Analysis of identified metabolites

The identified compounds refer to 11 HMDB chemical classes. Fatty acids[18 28 39] comprised the most prevalent chemical class, followed by amino acids[18 33] and phosphosphingolipids[17] (table 3).

A total of 5974 women were assessed for vitamin D status. The results were presented as total vitamin D,[32 35 37 40] although vitamin $D_2$, $D_3$ or 3-epi-25(OH)$D_3$[35] metabolites were measured. The results were stratified according to season of maternal sampling or latitude. Either <15 ng/mL (<37.5 nmol/L)[40] or <20 ng/mL (<50 nmol/L)[32 35 37] levels characterised vitamin D deficiency, but were statistically different in SGA pregnancies only in the first trimester.[32] Horgan et al[17] found a metabolite that could represent a vitamin D derivative, but it was only predictive in combination with 18 other compounds; this model had an area under the curve (AUC) of 0.90 (optimal OR, 44; 95% CI 9 to 214).[17]

The second most frequent targeted metabolite was homocysteine,[33 34] although levels were only differentiated between normal and SGA pregnancies when measured in second-trimester amniotic fluid, with a multiple linear regression model of $r^2$=0.012 and p=0.029.[33] Comparatively, the only common metabolite in the second-trimester maternal hair was margarate, with conflicting results since it was found to be either increased (AUC 0.72, 95% CI 0.58 to 0.86)[28] or decreased.[18] The N1,N12-diacetylspermine and the perfluorocarboxylic acids were associated with female SGA babies, not males. The former presented a fivefold decreased risk of SGA across

**Table 2** Subgroup analysis of included studies according to which metabolomics technique was applied

| Authors, year | Metabolomics technique | Maternal sample/storage temperature | Prediction model* | Targeted compounds | Coefficient of variation/limits of quantitation | Predictive compounds | Sensitivity /Specificity | AUC |
|---|---|---|---|---|---|---|---|---|
| **Nuclear magnetic resonance** | | | | | | | | |
| Luthra et al, 2018[38] | ¹H-NMR 1D NOESY with presaturation and homonuclear 2D J-resolved at 300 K Bruker 600 MHz Advance III HD spectrometer | Urine/−80°C | Targeted | Tyrosine, acetate, formate, trimethylamine | NA | None | | |
| **Gas chromatography coupled to mass spectrometry** | | | | | | | | |
| Costet et al, 2012[31] | GC-MS Simple headspace SPME-capillary GC | Urine/−20°C | Targeted | Trichloroacetic acid | <5%/0.01 mg/L | None | 0.1/0.93 | |
| Sulek et al, 2014[18] | GC-MS Thermo Trace GC Ultra system coupled to ISQ mass selective detector Capillary GC column: Phenomenex ZB-1701 (30 m × 250 µm id × 0.15 µm with 5 m guard column) | Hair/−20°C | Untargeted | NA | NA | ↓Lactate ↓Levulinate ↑2-methyloctadecanoate ↑Tyrosine ↓Margarate | | 0.998 |
| Delplancke et al, 2018[28] | GC-MS Agilent 7890B gas chromatograph, capillary column ZB-1701 (30 m × 250 µm id × 0.15 µm with 5 m guard column) 5977A mass spectrometer, electron impact ionisation | Hair/−20°C | Untargeted | NA | NA | ↑ Margaric acid ↑ Pentadecanoic acid ↑ Myristic acid‡ | | 0.72 0.73 0.73 |
| **Liquid chromatography coupled to mass spectrometry** | | | | | | | | |
| Grandone et al, 2006[33] | LC-MS/MS triple quadrupole Applera API 3000, TurbolonSpray ionisation | Amniotic fluid/−80°C | Targeted | Homocysteine | Unclear | ↑Homocysteine (1.29 µM; 1.05–1.51 µM) | | |
| Horgan et al, 2011[17] | UPLC-MS/MS Thermo Fisher LTQ Orbitrap, ESI | Plasma/−80°C | Untargeted | NA | NA | Hexacosanedioic acid, diglyceride, lyso-phosphocholine, sphinganine 1-phosphate, sphingosine 1-phosphate§ | | 0.90 |
| Ertl et al, 2012[32] | HPLC-MS/MS Shimadzu Prominence HPLC system with a column Phenomenex Luna C8 3×50mm; AbSciex API-5000 triple quadrupole, ESI | Serum/−80°C | Targeted | 25(OH)D₂; 25(OH)D₃ | 6.3%*, 6.6%† (D₂); 6.5%*, 7.3%† (D₃)/ unclear | ↓25,OH,vitamin D (12.16 ng/mL; 8.09–20.54 ng/mL) | 0.72/0.45 | |
| Gernand et al, 2013[40] | LC-MS/MS | Serum/−20°C | Targeted | 25(OH)D₂; 25(OH)D₃ | 8.2%* (D₂) 5.9%* (D₃)/<1 ng/mL | None | 0.39/0.66 | |

Continued

**Table 2** Continued

| Authors, year | Metabolomics technique | Maternal sample/storage temperature | Prediction model* | Targeted compounds | Coefficient of variation/limits of quantitation | Predictive compounds | Sensitivity /Specificity | AUC |
|---|---|---|---|---|---|---|---|---|
| Choi et al, 2016[34] | HPLC-MS/MS Waters HPLC system, Applied Biosystems API-4000 MS/ MS mass spectrometer | Serum/−20°C | Targeted | Methylmalonic acid; homocysteine | <10%*; <10%†/ unclear | None | | |
| Kiely et al, 2016[35] | UPLC-MS/MS Waters Acquity UPLC system, Waters Triple Quadrupole TQD mass spectrometer | Serum/−80°C | Targeted | $25(OH)D_2$; $25(OH)$ $D_3$; 3-epi-$25(OH)D_3$ | <6%*; <5%†/0.57 ng/mL ($D_2$); 0.26 ng/mL ($D_3$), 0.41 ng/mL (epi-$D_3$) | None | | |
| Ong et al, 2016[37] | LC-MS/MS Applied Biosystems Thermo Hypersil BDS C8 reverse-phase column | Plasma/unclear | Targeted | $25(OH)D_2$; $25(OH)$ $D_3$ | ≤10.3%*, †/<1.6 ng/ mL | None | | 0.12/0.87 |
| Wang et al, 2016[36] | LC-MS Agilent HPLC system, Applied Biosystems Sciex API-4000 triple quadrupole mass spectrometer | Serum/unclear | Targeted | PFOA; long-chain PFCA | 0.83–7.94%*; 1.57–24.7%†/0.07–0.45 ng/mL¶ | PFDeA (OR 3.14; 95% CI 1.07 to 9.19), PFUnDA (OR 1.83; 95% CI 1.01 to 3.32)** | | |
| Gong et al, 2018[41] | LC-MS/MS Shimadzu UK Limited UPLC system, ACE Excel 2 C18-PFP LC-column, Thermo Fisher Scientific Exactive Orbitrap mass spectrometer | Serum/unclear | Untargeted | NA | | ↑N1,N12-diacetylspermine** | | |
| Morillon et al, 2018[42] | UPLC-MS/MS Waters Acquity UPLC system, Waters Synapt G2-S mass spectrometer | Urine/−80°C | Untargeted | NA | | None | | |
| Others | | | | | | | | |
| van Eijsden et al, 2008[39] | GC-FID Solid phase extraction SPE, capillary GC | Plasma/−80°C | Semitargeted, lipid extraction | Elaidic, linoleic, alfa-linolenic, eicosatetraenoic, EPA, DPA, DHA DGLA, AA, adrenic, and Osbond acids | ≤2%–22%†/unclear | ↓ Eicosatetraenoic acid (OR 1.5; 95% CI 1.07 to 2.11), ↓DPA (OR 1.49; 95% CI 1.06 to 2.1) | | |

## Table 2 Continued

| Authors, year | Metabolomics technique | Maternal sample/storage temperature | Prediction model* | Targeted compounds | Coefficient of variation/limits of quantitation | Predictive compounds | Sensitivity /Specificity | AUC |
|---|---|---|---|---|---|---|---|---|

*Intra-assay coefficients of variation.

†Interassay coefficients of variation.

‡These metabolites were found in second-trimester hair segments.

§And more 14 metabolites that could not be identified certain based on chromatographic peak and mass: phenylacetylglutamine or formyl-N-acetyl-5-methroxykynurenamine; leucyl-leucyl-norleucine or sphingosine 1-phosphate; cervonyl carnitine and/or 1-alpha,25-dihydroxy-18-oxocholecalciferol; (15Z)-tetracosenoic acid or 10,13-dimethyl-11-docosyne-10,13-diol or trans-selacholeic acid; pencosenoic acid or cyclohexyl acetate or octanoic acid or methyl-heptenoic acid or 4-hydroxy-2-octenal or DL-2-aminooctanoic acid or 3-amino-octanoic acid; hydroxybutyrate or hydroxy-methylpropanoate or methyl methoxyacetate; lysophosphocoline and phosphocoline (more than 20 hits); phosphocoline or ubiquinone-8; acetylleucil-leucil-norleucinal or oleoylglycerone phosphate or LPA(0:0/18:2(9Z,12Z)) or 1-16:1lysoPE or phosphocoline(O-11:1(10E)/2:0) or (3s)-3,4-Di-N-hexanoyloxybutyl-1-phosphocoline or N-(3-hydroxy-propyl) arachidonoyl amine or N-methyl N-(2-hydroxy-ethyl) arachidonoyl amine or similar; lysophosphocholine (16:1) or cervoyl carnitine; preganediol-3-glucuronide or 3-alpha,20-alpha-dihydroxy-5-beta-pregnane-3-glucuronide; 6-hydroxyshingosine or (4OH,8Z,t18:1) sphingosine or 15-methyl-15-prostaglandin D2 or 15-R-prostaglandin E2 methyl ester.

¶Values for all studied metabolites.

**Predictive compounds only for female babies.

AA, arachidonic acid;AUC, area under the receiver operating characteristic curve; DGLA, dihomo-gamma-linolenic acid; DHA, docosahexaenoic acid; DPA, docosapentaenoic acid; EPA, eicosapentaenoic acid; ESI, electrospray ionisation; FID, flame ionisation detection; GC-MS, gas chromatography coupled to mass spectrometry;$^1$H-NMR, hydrogen nuclear magnetic resonance; HPLC, high performance liquid chromatography; LC-MS, liquid chromatography coupled to mass spectrometry; NA, not applicable; NOESY, nuclear Overhauser effect spectroscopy; PFCA, perfluorocarboxylic acid; PFDeA, perfluorodecanoic acid; PFOA, perfluorooctanoic acid; PFUnDA, perfluoroundecanoic acid; SPME, solid phase microextraction; UPLC, ultra-performance liquid chromatography.

quintiles. The perfluorodecanoic and perfluoroundecanoic acids presented OR of 3.14 (95% CI 1.07 to 9.19) and 1.83 (95% CI 1.01 to 3.32).[36] Tyrosine, an essential amino acid for infants, was part of the predictive model of maternal hair, combining five metabolites with an AUC of 0.998 (95% CI 0.992 to 1.0).[18] However, tyrosine did not predict SGA when urine samples were studied.[38] Methylmalonic acid,[34] acetate, formate or trimethylamine[38] did not differentiate SGA when compared with uncomplicated pregnancies (p>0.05).

### Risk of bias and applicability concerns

Figure 2 shows synthesised data for all included studies. See online supplementary material 3 for individual QUADAS-2 data.

Regarding the risk of bias, all cohort studies conducted a consecutive participant inclusion.[28 33–37 39] Nested case–controls matched cases and controls randomly[33–35 41] or according to maternal and infant characteristics.[17 18 38 42] One study[41] failed to mention matching procedures ('Patient Selection' domain). Researchers were not blinded to SGA status when interpreting metabolomics results,[17 18 28 32 35–41] and thresholds of targeted metabolites were not prespecified[31 33 36 38 39] ('Index Test' domain). Conversely, SGA identification was not influenced by the metabolomics test, although it was unclear when laboratory experiments were performed in some studies.[18 28 31 33 34 41] Birthweight charts were adequate, except for two studies. The first did not report which centile was chosen,[18] and the second used a centile designed for a different population[33] ('Reference Test' domain). Two studies were ranked as 'high risk' because not all participants were included in the analysis[31 37] ('Flow and Timing' domain).

The QUADAS-2 tool also highlights the importance of how the findings of the included studies are suitable to the review question. In the patient selection domain, it was ranked as 'high applicability concerns' when infants born between the 4th and the 10th centile, but with normal abdominal circumference growth velocity, were not included in the final analysis.[41] It was 'unclear' when the gestational age of maternal assessment was not standardised,[34] or was inferred by hair segment length,[28] or when few metabolites from untargeted studies were chosen for interpretation[41] ('Index Test' domain). Finally, it was 'high' when the birthweight charts applied did not correspond to the study population[18 33] ('Reference Standard' domain).

### Meta-analysis

From the 15 included studies, only 3 were designed for prediction purposes[17 18 42] and provided the AUC. The remaining reports described statistical differences of metabolites between SGA pregnancies and controls.[28 31–41] Accuracy measures were extracted when available (table 2). However, due to marked heterogeneity (tables 1 and 2) of gestational age at sampling, type of samples used, type of birthweight chart chosen, thresholds for vitamin D deficiency, metabolomics approach

**Table 3** Predictive metabolites summarised according to their chemical class, subclass and biological process

| Predictive metabolites | Chemical class | Chemical subclass | Metabolic pathway |
|---|---|---|---|
| Margarate | Fatty acyls | Fatty acids and conjugates | Lipid transport, metabolism, peroxidation |
| Pentadecanoic acid | Fatty acyls | Fatty acids and conjugates | Lipid transport, metabolism, peroxidation; fatty acid metabolism and biosynthesis |
| Myristic acid | Fatty acyls | Fatty acids and conjugates | Lipid transport, metabolism, peroxidation; fatty acid metabolism and biosynthesis |
| Eicosatetraenoic acid | Fatty acyls | Fatty acids and conjugates | Lipid transport, metabolism, peroxidation; lipid metabolism pathway |
| Docosapentaenoic acid | Fatty acyls | Fatty acids and conjugates | Lipid transport and metabolism, fatty acid metabolism, alpha linolenic acid and linoleic acid metabolisms |
| Tyrosine* | Carboxylic acids and derivatives | Amino acids, peptides and analogues | Catecholamine biosynthesis, phenylalanine and tyrosine metabolism, thyroid hormone synthesis, transcription and translation |
| Homocysteine | Carboxylic acids and derivatives | Amino acids, peptides and analogues | Glycine and serine metabolism, methionine metabolism |
| Hexacosanedioic acid | Carboxylic acids and derivatives | Dicarboxylic acid and derivatives | Fatty acid biosynthesis |
| Sphinganine 1-phosphate | Sphingolipids | Phosphosphingolipids | Sphingolipid signalling pathway, neuroactive ligand-receptor interaction |
| Sphingosine 1-phosphate | Sphingolipids | Phosphosphingolipids | Lipid metabolism pathway, sphingolipid metabolism |
| PFDeA | Alkyl halides | Alkyl fluorides | Not reported† |
| PFUnDA | Alkyl halides | Alkyl fluorides | Not reported† |
| 25,OH,vitamin D | Steroids and steroids derivatives | Vitamin D and derivatives | Lipid metabolism pathway |
| Diglyceride | Glycerolipids | Diradylglycerols | Adipocytokine signalling pathway |
| Lactate | Hydroxy acids and derivatives | Alpha hydroxy acids and derivatives | Gluconeogenesis, glycogenosis types IB and IC, pyruvate metabolism, triosephosphate isomerase |
| N1,N12-diacetylspermine | Carboximidic acids and derivatives | Carboximidic acids | |
| Lyso-phosphocholine | Glycerophospholipids | Glycerophosphocholines | Not reported† |
| 2-methyloctadecanoate | Saturated hydrocarbons | Alkanes | Not reported† |
| Levulinate | Keto acids and derivatives | Gamma-keto acids and derivatives | Not reported† |

*Essential amino acid for infants.
†No human metabolic pathways reported at KEGG.
KEGG, Kyoto Encyclopedia of Genes and Genomes; PFDeA, perfluorodecanoic acid; PFUnDA, perfluoroundecanoic acid.

and identified compounds, a meta-analysis could not be performed.

## DISCUSSION
### Main findings

In this first systematic review of metabolomics and adverse pregnancy endpoints, we presented techniques and metabolites which were studied for the prediction of SGA. Any effect on birth weight has important implications for perinatal research, since it is related to short-term and long-term outcomes,[43–46] and in different generations.[47 48] Intrauterine environment influences fetal growth through epigenetic processes: altered gene expression potentially leads to distinct phenotypes.[49] Metabolomics is the most adequate approach to study this outcome since it is most directly related to phenotype.[50]

Interpretation of metabolomics findings in pregnancy can be challenging. First, maternal metabolite concentrations are influenced by placental transfer to and from the fetus. The 'mirror effect', seen for maternal plasma and venous cord blood metabolites at birth,[51] cannot be ruled out when only maternal specimens are studied. Second, maternal exposure to distinct compounds may affect metabolite levels. Statistically significant differences

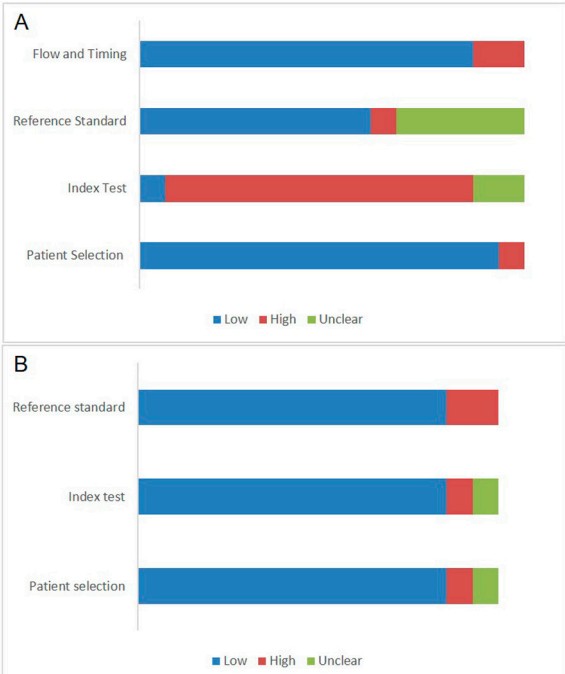

**Figure 2** Assessment of risk of bias (A) and applicability concerns (B) of individual studies.

between SGA infants and controls may not express the totality of underlying pathological pathways and have no clinical meaning. Finally, it is unclear when the processes leading to SGA are initiated. The disruption in maternal metabolism can theoretically occur at any time. In general the lower the gestational age at which the condition is suspected, the more severe the phenotype will be at birth.[52 53] Thus, the description of clinical data in translational studies must deal with all these confounding factors.

Gestational age at sampling is probably the most important parameter for prediction purposes. With timely prediction, women could be referred to specialised care and have increased surveillance, and this in turn may lead to a reduction in perinatal mortality. There are temporal changes in the maternal metabolome during pregnancy[28 54–57]; therefore, it is reasonable to expect distinctive metabolites at different stages of pregnancy, as reported here. Unfortunately, a wide or unclear definition of gestational age of sampling[34 36 38 40] renders a more precise interpretation impossible and may limit the clinical application of these results.

In contrast, gestational age at birth and birthweight centile seem to be the hallmarks of severity and prognosis of growth restriction.[6 58] Indeed, term and preterm SGA babies show distinct clinical phenotypes, and there are concerns that some babies <10th centile of birth weight are constitutionally small infants.[59–61] If only term deliveries are evaluated, the most severe cases of growth restriction may be potentially missed. Moreover, when term and preterm births are analysed together, or when lower cut-offs are not specified (eg, ≤3rd or ≤5th centile), the lack of predictive metabolites might mean that they are

distinct conditions. Thus, we hypothesise that the predictive performance of metabolomics may be improved if data are analysed by gestational age at delivery and by different cut-offs of birthweight centiles.

Evidence suggests that tobacco smoke has an impact on birth weight,[62–64] although it is uncertain how and when fetal growth is impaired. It is possibly related to oxidative stress,[65] and both maternal and fetal metabolism may be disturbed at delivery.[66 67] Studies that were included did not investigate cigarette-related chemicals or quantify exposure to tobacco smoke. Therefore, no relationship between SGA and tobacco was found. Hence, we suggest that tobacco interferes with ongoing metabolic pathological processes, or its disturbance is related to additional metabolic pathways other than the one examined by the included studies.

### Subgroup and metabolite findings
No reports have explored data on any maternal chronic condition, suspicion of SGA in pregnancy or number of fetuses. The lack of clear statements about participant selection has hindered data interpretation and precluded these analyses.

The majority of included studies performed a targeted approach, that is, a hypothesis-testing evaluation,[16 50] driven by epidemiological or experimental data regarding SGA newborns. None of the targeted metabolites[31–40] were in common with those found by 'hypothesis-generating' metabolic profiling[17 18 28 41 42] investigations. This reinforces the suggestion that various maternal metabolic pathways may be triggered by the SGA condition and be detected by different biological samples. However, since blood is a very complex sample and GC-MS only evaluates volatile molecules,[50] our findings may be biased by study methodologies.

Untargeted studies, as expected, have characterised several metabolites that may be validated in future investigations. Nine lipids and fatty acid metabolites,[17 18 28 39] two amino acids[18 33] and a steroid[17 32] have been identified as potential biomarkers of SGA.

All lipid-related metabolites identified are intermediates for energy storage and breakdown. Most metabolites were found in maternal blood[17] or hair of the SGA group.[18 28] Blood levels of saturated and monounsaturated non-esterified fatty acids apparently remain stable throughout pregnancy, while long-chain polyunsaturated fatty acid (docosahexaenoic acid and eicosapentaenoic acid, for example) measurements seem to show ethnicity-related changes.[57] Experimental data show the importance of hypoxia and oxidative stress to placental function, and ultimately to birth weight.[68 69] Findings from included studies may represent a dysregulation of lipid pathways at the placental level, but an association with maternal background is unclear. Therefore, we hypothesise that disorders of lipid metabolism may be the 'metabolic snapshot' of defective deep placentation[70] and might reflect maternal efforts to respond to impaired fetal growth.

Recommendations on the assessment of vitamin D and cut-offs to define vitamin D deficiency in pregnancy are controversial.[71] However, vitamin D supplementation decreases SGA risk.[72] In early pregnancy, vitamin D status has been related to SGA,[73 74] which is in accordance with this review, despite the inconsistent findings.[75] There is evidence that trophoblasts actively produce and secrete vitamin D metabolites,[76] but it is not clear how they mediate fetal growth impairment. Altered hepatic gene expression and liver function in vitamin D-deficient female rats[77] and single nucleotide polymorphisms[78] in vitamin D receptor gene have been suggested as mechanisms to be explored by a multidimensional omics approach.

Finally, homocysteine is an intermediate metabolite of the folate cycle. It is indirectly involved with DNA methylation and is a marker of folate deficiency.[79] Maternal levels rarely reach hyperhomocysteinaemia limits,[80] but folate depletion[81–83] and homocysteine itself[80] are thought to be associated with a higher SGA risk. In this review, homocysteine was only statistically different in SGA pregnancies when measured in amniotic fluid,[33] although within the normal ranges proposed for 17–21 weeks.[84] Since amniocentesis is generally performed in women at higher obstetrical risk, future studies should investigate whether homocysteine in amniotic fluid represents a confounding factor or a new biomarker.[85]

## Methodological quality

Most studies were ranked as 'low risk' of bias or applicability to the review question. However, the lack of clear descriptions of laboratory experiments, including sample preparation and storage, and blinding of the researchers to the case/control status are major pitfalls of the included studies.

## Strengths and limitations

To our knowledge, this is the first systematic review of metabolomics and an adverse pregnancy outcome (SGA). We presented possible biomarkers of SGA pathophysiology, metabolites implicated in lipid transport and metabolic pathways, as well as gluconeogenesis.

However, this analysis has some limitations. First, included studies showed heterogeneity, which is fundamental in systematic reviews. Indeed, there was a wide variety of participant characteristics and methods used, and not all authors provided a detailed description of methods employed. Although the Metabolomics Standards Initiative was released in 2007,[86] there is still poor adherence to guidelines.[87 88] Clear reporting[15 87 88] and data sharing in repositories are crucial steps in identifying features of interest, specifically possible biomarkers to be validated in the clinical studies.[15] Second, we could not perform a meta-analysis of the extracted data, impacting the translational potential of metabolomics.

Third, we considered that birth weight was a surrogate measure of intrauterine development. SGA and FGR are not interchangeable concepts. However, SGA has been used as a surrogate for FGR in many clinical studies due to difficulties in defining optimal intrauterine growth: (1) FGR diagnosis relies mostly on ultrasound measurements of fetal biometry,[3 89] which in turn is subject to systematic errors[90]; (2) intrauterine development is adaptive, rather than uniform[91] or only genetically driven[49]; and (3) growth impairment at birth better identifies adverse neonatal outcomes than during pregnancy.[58] It is recognised that changes in obstetric care occur when growth restriction is suspected, and neonatal outcomes are improved.[21 22] Thus, an accurate prediction of SGA during pregnancy will be a turning point in modern obstetrics.

## CONCLUSIONS AND IMPLICATIONS FOR PRACTICE

Using the available clinical tools, efforts to predict SGA remain disappointing. Since SGA is a heterogeneous condition, it benefits from metabolomics. This novel area of research allows analysis of numerous types of biological fluids and detects thousands of metabolites in complex samples.[15 16 25] However, findings of this systematic review must be interpreted with caution. The type of samples used may have influenced LC-MS (second-trimester maternal blood) and GC-MS (second-trimester maternal hair) findings in individual studies. Furthermore, the prediction of SGA in the context of maternal disorders, suspected FGR and twin pregnancies is an open field for future metabolomics studies, and environmental exposure investigation as well.

Surprisingly, none of the studies used ≤3rd centile of birth weight as a cut-off or analysed preterm deliveries and hypertensive syndromes. Considering our findings and the different phenotypic manifestations of SGA, we envision a better performance when (1) cut-offs other than the 10th centile are tested; (2) data on gestational age at sampling and at birth are standardised; and (3) other pregnancy-related syndromes are considered, especially hypertension. Thus, future metabolomics results should advance in these critical points.

Finally, all detected biomarkers were related to lipid pathways and energy metabolism. We consider that research efforts to predict SGA should focus on compounds involved in these pathways, up to the second trimester of pregnancy.

**Author affiliations**
[1]Department of Tocogynecology, Campinas' State University, Campinas, Brazil
[2]Department of Maternal and Child Health, Universidade Federal de Pernambuco, Recife, Pernambuco, Brazil
[3]Irish Centre for Fetal and Neonatal Translational Research (INFANT), University College Cork National University of Ireland, Cork, Ireland
[4]Clinics Hospital, Universidade Federal de Pernambuco, Recife, Brazil
[5]Obstetrics and Gynecology, Universidade Estadual de Campinas, Campinas, Brazil
[6]Department of Gynaecology and Obstetrics, St Thomas Hospital, Cork, UK
[7]Department of Epidemiology and Public Health, University College Cork, Cork, Ireland
[8]College of Medicine, University of Leicester, Leicester, UK
[9]Department of Women's and Children's Health, University of Liverpool School of Life Sciences, Liverpool, UK
[10]Obstetrics and Gynecology, University of Campinas, Campinas, Sao Paolo, Brazil

**Acknowledgements** We are grateful to Shauna Barret, from the Brookfield Library, University College Cork, Ireland, for her support with the literature search; Ting-Li Han, from the Department of Obstetrics, The First Affiliated Hospital of Chongqing Medical University, China, for providing additional data for this systematic review; and Luis Felipe D'Orsi, from the University of Campinas, for his support with methods' issues.

**Contributors** DFBL and A-CM have equally contributed to this report, and both are guarantors of this review. They elaborated on the protocol, searched the literature, selected studies, extracted data, assessed risk of bias and drafted the initial manuscript. RTS and EFMJ have participated in judging inclusion of studies, interpreting data and revising the manuscript. FPM has supported data extraction and has critically examined the clinical interpretation of the results. AK has discussed the quantitative data synthesis and supervised the report writing. PB, LCK and JGC have supervised and approved all steps. All authors have read and agree with this submission.

**Funding** DFBL (process number 88881.134512/2016-01) and RTS (88881.134095/2016-01) have scholarships awarded by Brazilian Federal Agency for Support and Evaluation of Graduate Education (CAPES). A-CM was granted a scholarship from Science Foundation Ireland for her doctoral thesis. PRETERM-SAMBA has granted sponsor from Brazilian National Research Council (CNPq) (Award 401636/2013-5) and from the Bill and Melinda Gates Foundation (grant OPP1107597), corresponding to the research call 'Grand Challenges Brazil: Reducing the burden of preterm birth', number 05/2013. This research received no specific grant from commercial or not-for-profit sectors. The sponsors have not intervened in the authors' decision to write the systematic review protocol or to submit this paper.

**Competing interests** None declared.

**Patient consent for publication** Not required.

**Provenance and peer review** Not commissioned; externally peer reviewed.

**Data availability statement** No data are available.

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
