## [Reviewer comments · BMJ Open]

ARTICLE DETAILS

TITLE (PROVISIONAL)	Examining the predictive accuracy of metabolomics for small for gestational age babies: a systematic review
AUTHORS	Leite, Debora; Morillon, Aude-Claire; Melo Júnior, Elias; Souza, Renato; McCarthy, F. P.; Khashan, Ali; Baker, Philip; Kenny, Louise; Cecatti, Jose

VERSION 1 - REVIEW

REVIEWER	Natalija Vedmedovska Rīga Stradiņš University, Latvia
REVIEW RETURNED	16-May-2019

GENERAL COMMENTS	The systematic review, titled "Examining the predictive accuracy of metabolomics for small for gestational age babies: a systematic review" is promoting the dissemination of knowledge on metabolomics and fetal growth restriction. The robust search strategies with data extraction were performed properly. Extensive discussion part was devoted to the role of metabolomics in FGR pathogenesis and potential implication for future research.
---

REVIEWER	Dr Alexandros A Moraitis University of Cambridge, UK
REVIEW RETURNED	21-May-2019

GENERAL COMMENTS	This is a very well designed and written paper. The literature search was done appropriately. The quality of control of the studies is correct. The discussion is quite long and if there is a word count problem I would suggest a slight reduction. Despite the lack of a meta-analysis due to the lack of enough data and the heterogeneity, I would highly recommend the publication of this paper.
---

VERSION 1 – AUTHOR RESPONSE

Firstly, in the Introduction, we apologize for not stating clearly our review question. We understand this real need; we have added it at Page 5, Line 24. Secondly, in the Methods section, we agree that public

and patients can be involved in the design and planning of a systematic review, and data should be shared when adequate. In this review, however, public and patients have not participated in any step, and there are no individual patient data (Page 9, Lines 15 and 18). Regarding literature search, we have written the full strategy for EMBASE electronic database (Page 6, Line 23). In this version, we have included a Supplementary Material with the detailed search strategy (Page 7, Line 5; Page 39, Line 23). In the Results section, we apologize for any errors regarding reference citation order. We have edited this data at Page 10 (Line 4).

In addition, there are some minor corrections of total word count (Page 2), the English language (Page 3), and order of Supplementary Materials (Pages 8; 10; 24). They are all marked; our aim is to improve readability and understanding. Finally, figures resolution were also amended; new files were uploaded.